

# Evaluation of a cadaveric wrist motion simulator using marker-based X-ray reconstruction of moving morphology

Joanna Glanville[1,2], Karl T. Bates[2], Daniel Brown[3], Daniel Potts[1], John Curran[1] and Sebastiano Fichera[1]

[1] School of Engineering, University of Liverpool, Liverpool, Merseyside, United Kingdom
[2] Department of Musculoskeletal & Ageing Science, University of Liverpool, Liverpool, Merseyside, United Kingdom
[3] Liverpool Orthopaedic and Trauma Service, Liverpool University Hospitals, Liverpool, Merseyside, United Kingdom

Corresponding author
Joanna Glanville,
hljglanv@liverpool.ac.uk

## ABSTRACT

Surgical intervention is a common option for the treatment of wrist joint arthritis and traumatic wrist injury. Whether this surgery is arthrodesis or a motion preserving procedure such as arthroplasty, wrist joint biomechanics are inevitably altered. To evaluate effects of surgery on parameters such as range of motion, efficiency and carpal kinematics, repeatable and controlled motion of cadaveric specimens is required. This study describes the development of a device that enables cadaveric wrist motion to be simulated before and after motion preserving surgery in a highly controlled manner. The simulator achieves joint motion through the application of predetermined displacements to the five major tendons of the wrist, and records tendon forces. A pilot experiment using six wrists aimed to evaluate its accuracy and reproducibility. Biplanar X-ray videoradiography (BPVR) and X-Ray Reconstruction of Moving Morphology (XROMM) were used to measure overall wrist angles before and after total wrist arthroplasty. The simulator was able to produce flexion, extension, radioulnar deviation, dart thrower's motion and circumduction within previously reported functional ranges of motion. Pre- and post-surgical wrist angles did not significantly differ. Intra-specimen motion trials were repeatable; root mean square errors between individual trials and average wrist angle and tendon force profiles were below 1° and 2 N respectively. Inter-specimen variation was higher, likely due to anatomical variation and lack of wrist position feedback. In conclusion, combining repeatable intra-specimen cadaveric motion simulation with BPVR and XROMM can be used to determine potential effects of motion preserving surgeries on wrist range of motion and biomechanics.

## INTRODUCTION

Surgical treatment at the end stages of wrist joint arthritis typically entails arthrodesis (fusion) (*Malarkey, Klena & Grandizio, 2022*), or a motion preserving surgery such as proximal row carpectomy, or more recently total wrist athroplasty (*Berber, Garagnani & Gidwani, 2018*). These surgeries substantially alter the biomechanics of the wrist. In

order to optimise their outcomes, detailed knowledge of the biomechanical changes they cause in comparison to the native wrist is important. Several studies have quantified parameters including wrist range of motion, carpal bone motion, centre of rotation and muscle moment arms in pre- and post-surgical wrists (*Sobczak et al., 2011*; *Debottis et al., 2013*; *Hooke et al., 2015*; *Nichols et al., 2017*; *Rust, Manojlovich & Wallace, 2018*; *Shiga et al., 2018*; *Fan et al., 2021*; *Saiz et al., 2021*). Due to the invasive nature of the surgeries and experimental techniques, studies are commonly performed *in vitro* using cadaveric specimens.

Artificial simulation of wrist motion is an essential element of many cadaveric studies. It has been achieved passively using traction weights (*Saiz et al., 2021*) and manual manipulation of the joint (*Hooke et al., 2015*; *Sobczak et al., 2011*; *Nichols et al., 2017*; *Stoesser et al., 2017*). Manual manipulation has the advantage of being simple to implement. Typically, a rod is fixed into the third metacarpal, which investigators use to move the wrist in desired motion patterns. A guide rail is sometimes used to help simulate repeatable flexion, extension (FE) and radioulnar deviation (RUD). The more complex dart thrower's motion (DTM), an oblique plane of movement from radial extension to ulnar flexion (*Moritomo et al., 2007*), and cirumduction tend to be simulated passively. Unfortunately it is difficult to achieve repeatable motion trials using manual manipulation, which makes experimental conditions between pre- and post-surgical wrists more difficult to control. Furthermore, all passive methods of motion simulation are limited in their representation of joint biomechanics because forces driving the simulated motions do not originate from muscles as they would in life.

To mitigate the disadvantages of passive methods, motion has also been simulated actively (*Erhart et al., 2012*; *Rust, Manojlovich & Wallace, 2018*; *Shiga et al., 2018*; *Fan et al., 2021*; *Werner et al., 1996*; *Iglesias, 2015*; *Shah & Kedgley, 2016*; *Razavian et al., 2022*). Typically, active wrist motion simulation is achieved by directing the distal tendons of wrist flexors and extensors to actuators that, using some form of control, produce the displacements and forces necessary to move the wrist. This method is advantageous in that the wrist motions are more repeatable than those achieved using passive simulation (*Iglesias, 2015*). It also allows better control of variables between surgical conditions as well as providing insight into the tendon forces necessary to achieve a motion.

As wrist motion comprises multiple muscles working synergistically, the extent to which an active simulator can represent *in vivo* motion is strongly influenced by the number of tendons included and the control strategy utilised. Relatively simple control can be achieved by applying pre-determined forces or weights to the five major wrist flexor and extensor tendons in order to simulate FE, RUD and even DTM (*Erhart et al., 2012*; *Rust, Manojlovich & Wallace, 2018*). More complex simulators use position feedback whereby errors between actual and desired wrist angles are minimised (*Werner et al., 1996*; *Iglesias, 2015*; *Shah & Kedgley, 2016*). Position feedback allows user control over wrist angle and as such, simulators that implement this strategy have achieved circumduction and highly specific DTM planes (*Werner et al., 2004*). However, as most simulators actuate only five or six tendons, they cannot fully recreate muscle actions at the wrist. To better represent the native wrist, one recent simulator included eleven muscles which were controlled by

an optimisation algorithm that minimises tendon forces (*Razavian et al., 2022*). Though, without the availability of *in vivo* force data for comparison, it is still important to analyse and interpret the cadaveric data with limitations of the simulator in mind.

An additional challenge for both *in vitro* and *in vivo* wrist motion studies is accurately capturing the precise 3D motion of carpal bones. The current gold-standard solution for *in vivo* studies is 4D CT scanning (*Brinkhorst et al., 2020*). However, when comparing wrist biomechanics between surgical conditions, cadaveric studies are much more feasible. Such studies commonly track bones using optical or electromagnetic motion capture systems, whereby markers or sensors are attached rigidly to bones *via* rods (*Werner et al., 2011*; *Stoesser et al., 2017*; *Shiga et al., 2018*). While effective when tracking larger bones, it seems likely that these rods introduce unnatural torques to the small carpal bones which may alter their kinematics. An alternative option is biplanar X-ray videoradiography (BPVR). When combined with marker-based X-ray Reconstruction of Moving Morphology (XROMM), this technique allows 3D motion reconstruction based on markers implanted directly into the bones and has the potential for sub-millimetre and sub-degree accuracy (*Brainerd et al., 2010*). *In vivo* wrist angles have been successfully reconstructed using a markerless variation of XROMM (*Akhbari et al., 2019b*; *Akhbari et al., 2020*). However, as carpal bones are difficult to demarcate in biplanar X-ray images, marker-based XROMM is likely to yield more accurate carpal kinematics. The current prevailing method by which to animate joints in marker-based studies is manual manipulation (*Manafzadeh, 2020*). This is subject to the limitations of passive joint manipulation and radiation exposure to the investigators. In the present study we aim to demonstrate the feasibility of using active joint motion simulation in a marker-based XROMM study.

This article will describe the development of our active wrist motion simulator. Its design is based upon previous simulators that actuate five to six tendons (*Erhart et al., 2012*; *Werner et al., 1996*; *Iglesias, 2015*; *Shah & Kedgley, 2016*). We will also report the results of a pilot experiment that aimed to evaluate the simulator against three principal criteria. First, the simulator should be able to produce planar and complex wrist motions in a functional range. Second, control measures should ensure that meaningful comparisons can be drawn between pre- and post- surgical conditions. Third, the simulator should be compatible with BPVR, which will facilitate the future study of detailed kinematic parameters.

## MATERIALS AND METHODS

The motion simulator was designed to facilitate the comparative study of wrist kinematics before and after a cadaveric surgery. Design criteria were established to this end. Foremost, functional wrist angles should be reached in FE, RUD, DTM and circumduction, through the application of prescribed displacements to five main wrist flexor and extensor tendons. To allow valid comparison between pre- and post-surgical wrists, tendon displacements should be repeatable across all wrist conditions. Additionally, intra-condition motion trials should have repeatable wrist angles and tendon forces.
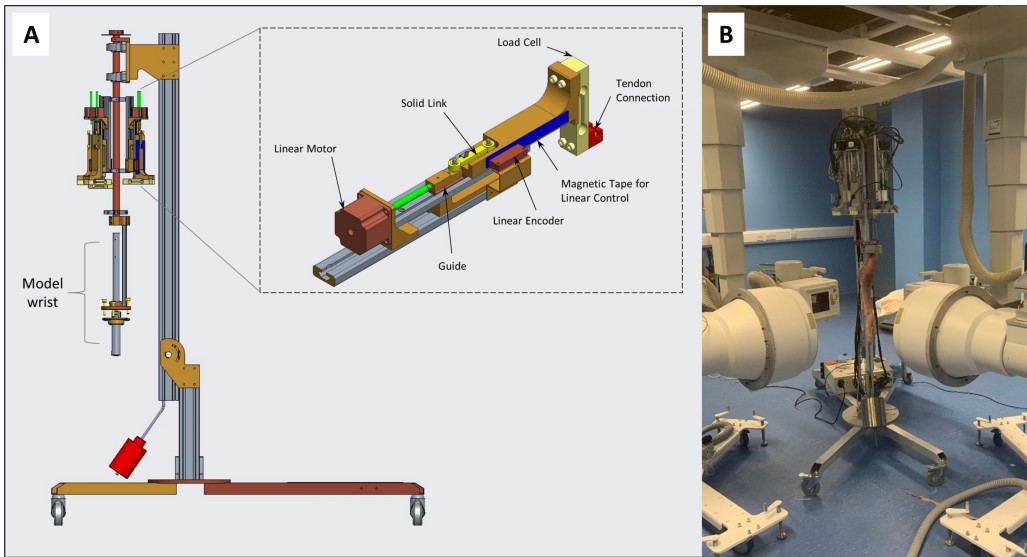

**Figure 1 Motion simulator and biplanar X-ray set up.** (A) Computer Aided Design drawing showing the main elements of the wrist motion simulator with a model arm mounted. Five identical units are positioned around a central rotating axis. The arm is mounted below this assembly *via* a bracket, metal rod, clamps and pins. The zoomed in panel depicts one unit designed to actuate a single tendon. (B) The whole motion simulator, the main elements are suspended vertically using an aluminium frame. The wrist is positioned inside the biplanar X-ray capture volume.

## Simulator hardware

The simulator comprises five identical units with which to actuate the tendons of a human forelimb specimen (Fig. 1A). The following description applies to one such unit. A non-captive linear actuator (Reliance Precision Ltd, Huddersfield, UK) was used to drive tendon motion. 80 mm of actuator travel was allowed based on *in vitro* measurements of tendon displacement during FE and RUD (*Tang, Ryu & Kish, 1997*; *Horii, An & Linscheid, 1993*; *Tang et al., 1999*). The linear actuator was connected in series to a metal rod whose travel was measured by a Magnetic Linear Encoder (Machine-Dro, Hoddesdon, UK) as a proxy for tendon displacement. 80 mm of encoder travel was allowed. Attached to the end of the rod was a Single Point Load Cell (Omega, Manchester, UK) that measures the resultant force applied to the tendons. The load cell has a capacity of 150 N as previous simulators have reported tendon forces up to 100 N (*Dimitris et al., 2015*). These units are positioned symmetrically around the central axis such that tendons can be attached differently according to whether a left or right arm is being used (Table 1). The whole assembly is suspended vertically such that the wrist hangs in free space with no metal obstruction (Fig. 1B), though a hinge joint at the base allows the assembly to be oriented horizontally as well. The sensors and actuators are controlled *via* a PIC32 micro-controller.

## Motion simulator software and control

Custom software is used to control the simulator. There are three control features. The first equalises force across all tendons and is implemented before the commencement of a new motion. The second mediates actuator displacements during motion simulation. The

**Table 1** Tendons and the axes to which they are connected for left and right arms along with their displacement allowance and maximum force.

| | Tendon | | | | |
|---|---|---|---|---|---|
| | *ECRL* | *ECRB* | *ECU* | *FCR* | *FCU* |
| *Axis (Left Arm)* | 1 | 2 | 3 | 5 | 4 |
| *Axis (Right Arm)* | 3 | 2 | 1 | 4 | 5 |
| *Actuator and encoder travel (mm)* | 80 | 80 | 80 | 80 | 80 |
| *Maximum tendon force (N)* | 80 | 86 | 74 | 125 | 211 |

Notes.
ECRB, Extensor carpi radialis brevis; ECRL, Extensor carpi radialis longus; ECU, Extensor carpi ulnaris; FCR, Flexor carpi radialis; FCU, Flexor carpi ulnaris.

third sets maximum force boundaries for each tendon and is also active during motion simulation.

1. The force control feature allows force to be equalised across all tendons to a user specified magnitude. The feature uses a feedback control loop where error is defined as the difference between desired and actual tendon forces. Actuators are re-positioned after each loop cycle until the error approaches zero. Once the wrist is positioned neutrally in the simulator, this control feature is applied in order to set the actuators to a zero position for each tendon.

2. Actuator displacement control allows users to specify the movement of each actuator. Actuator position profiles specify the displacement of each actuator away from its zero position at a sampling frequency of 50 Hz. A feedback loop between desired and actual actuators displacement ensures that the prescribed motion profiles are correctly executed. This method of control allows the same actuator displacements to be applied to both pre- and post-surgical wrists.

3. The maximum force boundary ensures that tendon forces do not exceed their total physiological muscle forces. These were estimated for each muscle at a specific tension of 32 N/cm2 (*de Monsabert et al., 2017*) using physiological cross sectional areas (PCSAs) derived from muscle volumes of young healthy subjects (*Holzbaur et al., 2007*) (Table 1).

## Simulating a motion
### Specimen mounting
Three rigid external fixation pins are fixed into the radius and humerus, which are in turn fixed to an aluminium mounting bar using adjustable clamps. The mounting bar is bolted to a bracket at the bottom of the central axis of the simulator such that the limb runs in line with the axis and the wrist hangs naturally with no exact neutral position imposed. To remove an arm for surgery the mounting bar is unbolted from the bracket. The pins and clamps remain in place during surgery before the bar is remounted. This allows the specimen to be reinstalled into the simulator in close to the same configuration. *Extensor carpi radialis longus* (ECRL), *Extensor carpi radialis brevis* (ECRB), *Extensor carpi ulnaris* (ECU), *Flexor carpi radialis* (FCR) and *Flexor carpi ulnaris* (FCU) tendons are connected to each unit *via* polyethylene sutures attached between musculotendinous junctions distally

and load cells proximally. The tendons of left and right arms are mounted as mirror images to each other (Table 1).

### Generating actuator displacement profiles

Actuator displacement patterns necessary to produce specific motion patterns were created artificially using measured tendon excursions of a model wrist. The model consisted of two aluminium rods connected by a universal joint. Kevlar 'tendons' were attached at the base of the distal segment mimicking the insertions of the five wrist flexors and extensors. The model was connected to the simulator as described above. All tendons were loaded equally to 2 N before the rigid links between the load cells and actuators (Fig. 1A) were replaced by springs. All actuators were jogged backwards six cm in order to load the tendons. With spring loaded tendons, the model wrist could be manually manipulated and resulting 'tendon' displacements were recorded using the linear encoders. Using this method, actuator displacement profiles were generated for flexion, extension, radial deviation (RD), ulnar deviation (UD), radial extension (RE), ulnar flexion (UF) and circumduction. While recording actuator displacement profiles, the simulator was oriented horizontally so that gravity did not influence spring displacement.

These recorded profiles acted as a template for the actuator displacements applied to the cadaveric specimens. A simple script was developed that allowed individual displacement profiles to be scaled to produce different magnitudes of wrist motion. All displacement profiles were designed begin and end at 0 mm of actuator displacement, with the exception of circumduction which returned to an extension configuration. Due to profile length limitations, circumduction was split into two profiles. The first profile comprised extension-UD-flexion, and the second flexion-RD-extension. Circumduction data was joined during data analysis.

### Motion simulation and data acquisition

On mounting a cadaveric specimen to the simulator, all tendon forces were equalised to 2 N; the same force used to generate the position profiles. In this position all encoder readings were set to zero. Desired actuator displacement profiles were applied and the resulting motion observed. If necessary, displacement profiles were scaled in order to achieve a more optimal or different magnitude of wrist motion. Range of motion was qualitatively assessed by investigators prior to data collection and later verified using BPVR images. The data output from a motion trial comprises tendon forces and actuator displacements at a sampling frequency of 50 Hz.

## Experimental procedure

### Specimen acquisition, usage and storage

Data was collected from three right and five left cadaveric wrists originating from eight donors aged 67–93. Specimens extended from distal humerus to the fingertips. They were obtained from the National Repository and stored at the University of Liverpool. Computed tomography (CT) scans were performed at the Royal Liverpool University Hospital. All three organisations hold appropriate HTA licences. Written informed consent was received from all subjects before donating their bodies to medical science. Ethical approval for the

study was granted by the University of Liverpool Central University Research Ethics Committee (Reference 8009). Data collection was carried out after one freeze-thaw cycle. It has been previously determined that the biomechanical properties of ligaments, tendons and articular cartilage are not significantly altered after less than three freeze thaw cycles (*Moon et al., 2006*; *Huang et al., 2011*; *Peters et al., 2017*). Two specimens were excluded from the analysis, one due to midcarpal instability and the other due to improper placement of a total wrist replacement prosthetic.

Specimens were stored at −20 °C prior to thawing at room temperature for 24 h. For each wrist, data was collected over two consecutive days with overnight storage at 4 °C. On the first day, surgery was performed to implant radio-opaque beads into bones, suture tendons and fix pins into the radius and humerus. The wrist was CT scanned before simulated motion trials were captured using BPVR. Total Wrist Replacement surgery was performed at the end of the day. During day two, data collection protocols were repeated on the post-surgical wrist. The wrist was then re-frozen and CT scanned at a later date. Prior to any data collection, anatomical normalcy of the wrist was confirmed by clinicians through X-ray and intra-operative assessment.

### Surgical preparation

The cadaveric wrist underwent two rounds of surgery, both of which were performed by the clinical author who is an experienced wrist arthroplasty surgeon. Care was taken to ensure ligaments and tendons remained intact. For the first surgery, a longitudinal skin incision was made dorsally from the distal radius to the third metacarpal base. The wrist joint capsule was opened using a standard distally based flap to expose the proximal carpal row. One millimeter tantalum beads (X-medics, Frederiksberg, Denmark) were implanted into dorsal aspects of the radius, ulna, second metacarpal (MCII) and third metacarpal (MCIII). Holes, two mm in diameter, were drilled through bone cortices using a surgical drill (Acumed, Hillsboro, OR, USA). The holes were filled with a gel based superglue and a single bead inserted into each. Three beads per bone were implanted. After the superglue was confirmed to have cured, the wrist joint capsule was sutured closed.

Tendons of ECRL, ECRB, ECU, FCR and FCU were isolated at their distal musculotendinous junctions. The latter two were accessed *via* volar skin incisions. Braided 1.3 mm polyethylene sutures (Arthrex, Naples, FL, USA) were secured to each tendon using a multiply cross-locked cruciate stitch based upon the Adelaide tendon repair technique (*Jordan et al., 2015*). To preserve muscle line of action, the sutures were guided proximally adjacent to their respective muscle and exited *via* a skin incision at either the medial or lateral epicondyle of the humerus according to the muscle's origin. Silicon tubing was inserted around each length of suture to reduce friction against soft tissues. The dorsal and volar skin incisions were sutured closed. Finally, holes were drilled through the radial aspects of the radial and humeral cortices; two in the radius, one in the humerus, to allow five mm surgical half pins (Smith & Nephew, Watford, UK) to be inserted approximately parallel to each other.

The second surgery was a total wrist replacement using the Motec prosthesis (Swemac, Linköping, Sweden). Surgery was performed as per the manufacture's guidelines (*Reigstad et al., 2011*), though intra-operative radiography was not available.

### CT scanning

Two CT scans were taken of each wrist, one post-bead implantation and one post-total wrist replacement, using a SOMATOM scanner (Siemens Healthcare, Erlangen, Germany). Scans extended from mid-forearm to metacarpal heads and CT settings were 120 kVp, 76 mA, slice thickness = 0.6 mm, voxel size = 0.27 mm2.

### Motion capture

Two rounds of biomechanical data collection were performed for each wrist. The first used the anatomical wrist while the second was performed following the total wrist replacement. All motion capture was performed using the biplanar X-ray imaging facility at the University of Liverpool. The system, supplied by Imaging Systems & Services Inc (Painesville, OH, USA), comprises two X-ray tubes, two EPS 45–80 High Voltage X-ray Generators (EMD Technologies, Saint-Eustache, Quebec, Canada), two image intensifiers (IIs) and two Phantom Miro M120 digital video cameras (Vision Research, Wayne, NJ, USA). The X-ray tube and II pairs were positioned orthogonally with source to image distances of 110 cm. The first round of data collection used a peak X-ray generator voltage of 60 kVp and an average tube current of 80 mA, while the second used 80 kV and 32.0 mA, which allowed bead visualisation through the wrist prosthesis. Image undistortion and 3D-volume calibration were performed before and after both rounds of data collection as described by *Brainerd et al. (2010)*. A perforated metal undistortion sheet (Part 9255T641, McMaster-Carr, Robinson, NJ, USA) and a calibration cube containing 64 tantalum beads of known spacing were used.

The wrist joint was positioned inside the biplanar capture volume at an angle of approximately 45 degrees to both X-ray tubes (Fig. 1B). Seven different motion patterns were simulated using the motion simulator. These were flexion, extension, RD, UD, RE, UF and Circumduction. Each motion was cycled twice before five trials per motion were recorded with BPVR. A static trial with the wrist in its neutral position was taken for each motion pattern after the fifth motion trial. Prior to each new motion, tendon load was set to 2 N. Order of the motion trials was randomised between specimens. Image capture was at 50 fps for all trials apart from the longer circumduction trials which were captured at 30 fps. To synchronise X-ray acquisition with motion simulation an active high TTL signal was sent to the X-ray using an Arduino Uno (Arduino, New York, NY, USA), which triggered X-ray capture at the point of simulated motion commencement.

### X-ray reconstruction of moving morphology

XROMM protocols were followed to produce 3D reconstructions of motion trials (*Brainerd et al., 2010*). 3D polygonal meshes of the radius, ulna, MCII, MCIII, tantalum beads and wrist prosthesis components were segmented from the CT scans using Mimics 25.0 software (Materialise NV, Leuven, Belgium). Tantalum bead centroids were determined in the global CT coordinate system using the XROMM specific tool shelf (*MayaTools, 2023*) in Maya

2023 animation software (Autodesk, Mill Valley, CA, USA). Using XMA-lab software (*Knorlein et al., 2016*), tantalum beads were tracked in both videos across all frames of motion. Bead locations were reconstructed in 3D and separated into rigid body groups according to the bone into which they were inserted (*Knorlein et al., 2016*). XMA-lab bead locations were lined up with the CT scan calculated bead centroids. Rigid body transformations (RBTs) of each bone between each frame of video were calculated in the global CT coordinate system (*Knorlein et al., 2016*). The RBTs were smoothed using a low-pass Butterworth filter with a 2 Hz cut-off frequency, chosen as per the XMALab guidelines. In Maya the RBTs were applied to their corresponding bone model to create the final motion trial reconstruction.

### Calculating wrist motions

An anatomical radial coordinate system (RCS) was placed in the radius in accordance with the modification to the International Society of Biomechanics' recommendation (*Wu et al., 2005*) described by *Akhbari et al. (2019a)*. Briefly, the $x$-axis was aligned with the distal radial shaft and the origin defined as the point at which the $x$-axis intersects the distal articular surface. The $y$-axis is a line perpendicular to the $x$-axis crossing the origin and the centre of the sigmoid notch. The $z$-axis is the cross product of the $x$-axis and $y$-axis. Rotation about the $x$-axis corresponds to pronation (+)/supination (-), the $y$-axis to flexion(+)/extension (-) and the $z$-axis to ulnar (+)/radial (-) deviation. A third metacarpal coordinate system (MCS) was placed according to *Akhbari et al. (2020)*. The $x$-axis aligned with the diaphysis, the $y$-axis was a best fit line between the centroids of MCII, MCIII and MCIV and the $z$-axis was a cross product of $x$ and $y$ axes. To ensure consistent coordinate system placement before and after surgery, the bone meshes were aligned using N-Point and Global Registration tools in Mimics 25.0. The anatomically defined coordinate systems were applied to the total wrist replacement bones.

Overall wrist motion was defined according to the rotations of MCS about the RCS with respect to a neutral wrist position defined as the alignment of RCS and MCS. As wrists were unlikely to have been mounted in exact neutral, initial wrist positions were extracted from the static trials. The average profiles of X, Y and Z axis rotations for each motion were calculated by truncating four of the five trials to match the length of the shortest trial and then taking the mean angle at every sampling point. With the exception of circumduction, mean motion profiles were further truncated so as to end at the maximum joint rotation. For flexion and extension trials the maximum was taken as the largest rotations about the $y$-axis, for RD and UD it was the largest rotations about the $z$-axis and for RE and UF it was the largest rotations about the $y$-axis and $z$-axis combined.

The mean motion profiles of each subject were visualised by plotting RUD angles on the $x$-axis and FE angles on the $y$-axis. In order to determine whether a functional wrist ROM had been reached, the maximum in-plane flexion, extension, RD and UD were extracted. Out-of-plane RUD at the magnitudes of flexion/extension and out-of-plane FE at the magnitudes of RD/UD were also extracted as these were not eliminated by the simulator control mechanism.

Since the DTM plane occurs at different angles depending on the type of movement being performed (*Palmer et al., 1985*), orientations of simulated RE and UF planes were determined using the method described by *Brigstocke et al. (2014)*. A linear regression model was applied to the plotted RE and UF profiles. Models with $R^2$ values lower than 0.7 were excluded (*Brigstocke et al., 2014*). Angles of the regression lines relative to the sagittal axis of the forearm were used to approximate the orientation of RE and UF motions. RMSE between regression lines and DTM profiles were calculated as a measure of the model's fit.

In order to quantitatively compare the simulated circumduction envelopes with previous studies, an ellipse was fitted to each envelope and its area and major axis orientation was extracted. As measure of ellipse fit, RMSE was calculated between the magnitudes of 100 vectors on the ellipse and the corresponding vectors on the circumduction envelope.

Statistical comparisons of pre- and post-surgical initial and maximum wrist angles as well as DTM and circumduction parameters were performed in R4.3.2. Paired samples $t$-tests or Wilcoxon tests were used depending on whether Shapiro–Wilks tests identified a normal distribution.

### Calculating repeatability

RMSE was used to represent the inter-trial and inter-specimen repeatability of tendon forces, actuator displacements and wrist angles. For inter-trial RMSE, mean force, displacement and wrist angle profiles were calculated in each specimen for each tendon/rotation axis. RMSEs of individual trial profiles against specimen mean profiles were then calculated, and the maximum trial RMSE identified for each specimen. For inter-specimen RMSE, the specimen mean profiles were interpolated to 100 data points (200 for circumduction) and an overall mean calculated. RMSEs of specimen mean profiles against the overall mean were calculated. To verify repeatability of actuator displacements across different wrist conditions, RMSE calculations were performed for the anatomical and total wrist replacement conditions combined. As tendon force and wrist angles were expected to vary across conditions, error calculations were performed separately for anatomical and total wrist replacement conditions. Mean inter-trial and inter-specimen RMSEs were taken for each motion. All calculations were performed in MATLAB (R2022b; MathWorks Inc., Natick, MA, USA).

## RESULTS

### Wrist angles

The motion simulator was able to produce recognisable motion trials for the four major planar wrist motions. Across the six specimens, maximum wrist angles for flexion, extension, RD and UD were respectively 50.5 ± 6.3°, 57.2 ± 7.9°, 20.9 ± 8.3° and 29.2 ± 4.6° (Table 2). All planar motions were accompanied by out of plane motions (Table 2). Wrists tended to ulnarly deviate during flexion. The total wrist replacements achieved similar maximal angles during Flexion (49.1 ± 7.6°), Extension (57.3 ± 13.4°), RD (22.1 ± 13.3°) and UD (26.0 ± 7.4°). Initial wrist positions did not vary significantly between surgical conditions (Table S1).

**Table 2** **The maximum in and out of plane wrist angles (°) achieved during flexion (+), extension (-), radial deviation (-) and ulnar deviation (+) before and after a total wrist replacement.** Out of plane wrist angles are reported at the point of maximum excursion in the desired plane. All angles represent mean ± standard deviation, and statistically significant differences between anatomical and Motec wrists are indicated by a $p$-value < 0.05 ($n = 6$).

|  |  | Flexion | Extension | Radial deviation | Ulnar deviation |
|---|---|---|---|---|---|
| In plane | *Anatomical wrist* | $50.5 \pm 6.3$ | $-57.2 \pm 7.9$ | $-20.9 \pm 8.3$ | $29.2 \pm 4.6$ |
|  | *Motec wrist* | $49.1 \pm 7.6$ | $-57.3 \pm 13.4$ | $-22.1 \pm 13.3$ | $26.0 \pm 7.4$ |
|  | *P-value* | 0.736 | 0.985 | 0.772 | 0.157 |
| Out of plane | *Anatomical wrist* | $8.2 \pm 3.9$ | $-1.4 \pm 6.6$ | $5.8 \pm 7.5$ | $-2.4 \pm 9.6$ |
|  | *Motec wrist* | $8.6 \pm 5.7$ | $-6.4 \pm 10.7$ | $-1.7 \pm 11.5$ | $-5.6 \pm 13.9$ |
|  | *P-value* | 0.861 | 0.009 | 0.485 | 0.818 |

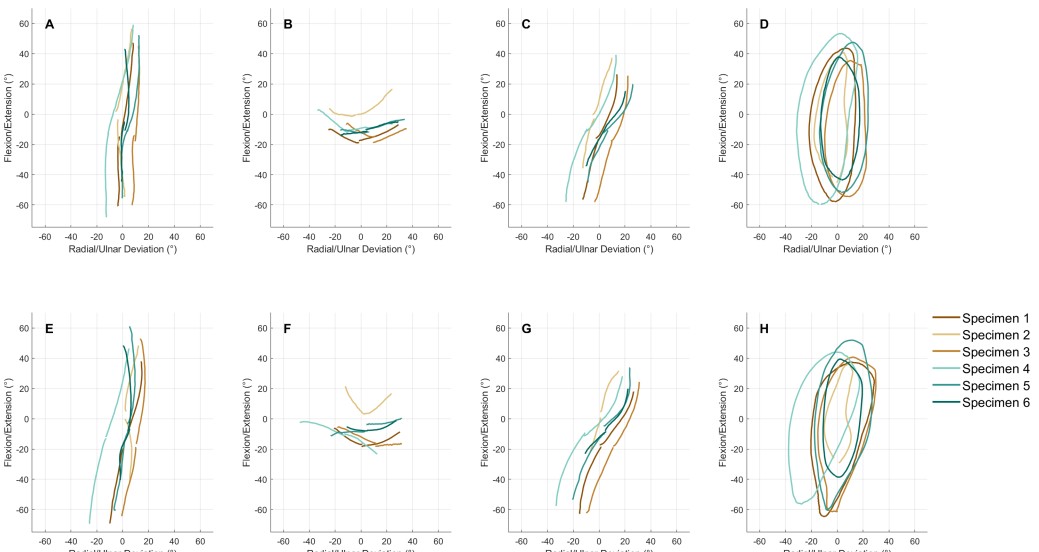

**Figure 2** **Mean motion paths of six wrists during flexion/extension (A&E), radioulnar deviation (B&F), dart thrower's motion (C&G) and circumduction (D&H).** Motion paths of the third metacarpal with respect to the radius are represented by rotations (°) about the flexion(+)/extension(-) axis plotted against the radial(-)/ulnar(+) deviation axis of the radial coordinate system. (A–D) Motion paths of the anatomical wrist. (E–H) Motion paths of the Motec wrist. Each colour corresponds to a single specimen.

DTM motion profiles are visualised in Figs. 2C & 2G. In the anatomical wrists, RE planes were oriented 20.4 ± 6.1° and UF planes 26.2 ± 8.1° from the sagittal plane. Mean RMSEs between regression lines and motion profiles were 0.63 ± 0.43° and 1.51 ± 0.62° for RE and UF respectively. After total wrist replacement surgery, orientation of the DTM planes increased, but these increases were not statistically significant. Orientation of the RE plane increased to 27.5 ± 11.0° ($p = 0.132$) and the UF plane to 31.4 ± 6.0° ($p = 0.211$).

Circumduction was successfully simulated in the anatomical wrists, though there was a notable decrease in ulnar deviation as the wrist moved from extension to flexion, which occurred in three of the six specimens Figs. 2D & 2H. Imperfections in the circumduction envelopes are reflected by the relatively high mean RMSEs of their fitted ellipses which were

$2.2 \pm 1.3°$ and $3.6 \pm 1.3°$ for the anatomical wrists and total wrist replacements (Figs. 2D & 2H). Inter-specimen circumduction ellipse areas were variable across the six specimens (Figs. 2D & 2H), though mean ellipse areas of $2{,}608 \pm 771°°$ and $2{,}847 \pm 1{,}118°°$ were similar in pre- and post- surgical wrists ($p = 0.699$). Orientation of circumduction ellipses tended to increase after the total wrist replacement surgery. In the anatomical wrist the ellipse was oriented $4.2 \pm 2.8°$ from the sagittal plane, this increased to $11.6 \pm 5.5°$ after surgery ($p = 0.007$).

### Repeatability

Inter-trial repeatability of the motion simulator is shown in Fig. 3, which presents mean (standard deviation) wrist angles, tendon forces and actuator displacements during the radial extension trials of one specimen (see Fig. S1 for further motions and specimens). Mean inter-trial RMSEs of wrist angle and tendon force trials were below 1° and 2 N across all motions in both anatomical wrists and total wrist replacements (Table 3). Mean wrist angle error ranged between 0.20° and 0.68°. Mean tendon force error ranged between 0.23 N and 1.41 N. Tendon displacement error ranged between 0.07 mm and 0.25 mm. The largest inter-trial variation tends to occur in tendon forces, particularly in the tendon to which the largest force is applied. For example, during radial extension, ECRL experiences the largest forces and has the largest variation between trials (Fig. 3). These patterns are true for both anatomical and total wrist replacement conditions (see Fig. S1).

Mean inter-specimen RMSEs for wrist angle and tendon forces were as large as 12.97° and 9.29 N respectively (Table 4). The largest inter-specimen wrist angle variation tended to occur about the $Y$-axis, while inter-specimen tendon forces varied most in agonist tendons (Tables S2 & S3).

## DISCUSSION

In the present study we have developed an active wrist motion simulator that is capable of producing sufficient FE, RUD, DTM and circumduction ranges required to perform activities of daily living (ADLs). Our simulator allows pre- and post-surgical comparisons by controlling tendon displacements and arm mounting position, allowing range of motion and artificial tendon forces to be compared across surgical conditions. Compatibility with BPVR and highly accurate marker-based XROMM techniques (Brainerd et al., 2010) will allow detailed study of in vitro wrist kinematics in future studies.

### Wrist angles

Functional range of motion (fROM) refers to the maximum angles of FE and RUD required to perform ADLs (Palmer et al., 1985). Since a goal of many wrist surgeries is to preserve ADLs, it is important that simulated motions are within this functional range. Across the six wrists used in this study, mean flexion was 50.5°, extension was 57.2°, RD was 20.9° and UD was 29.2° (Table 2). This corresponds favourably with previous measures of fROM which find that a functional FE arc is between 35–94°, and an RUD arc between 20–57° (Brigstocke et al., 2013; Dogan et al., 2019; Gracia-Ibanez et al., 2020; Palmer et al., 1985; Ryu et al., 1991). Variation in previous fROM measures may be due to the differing
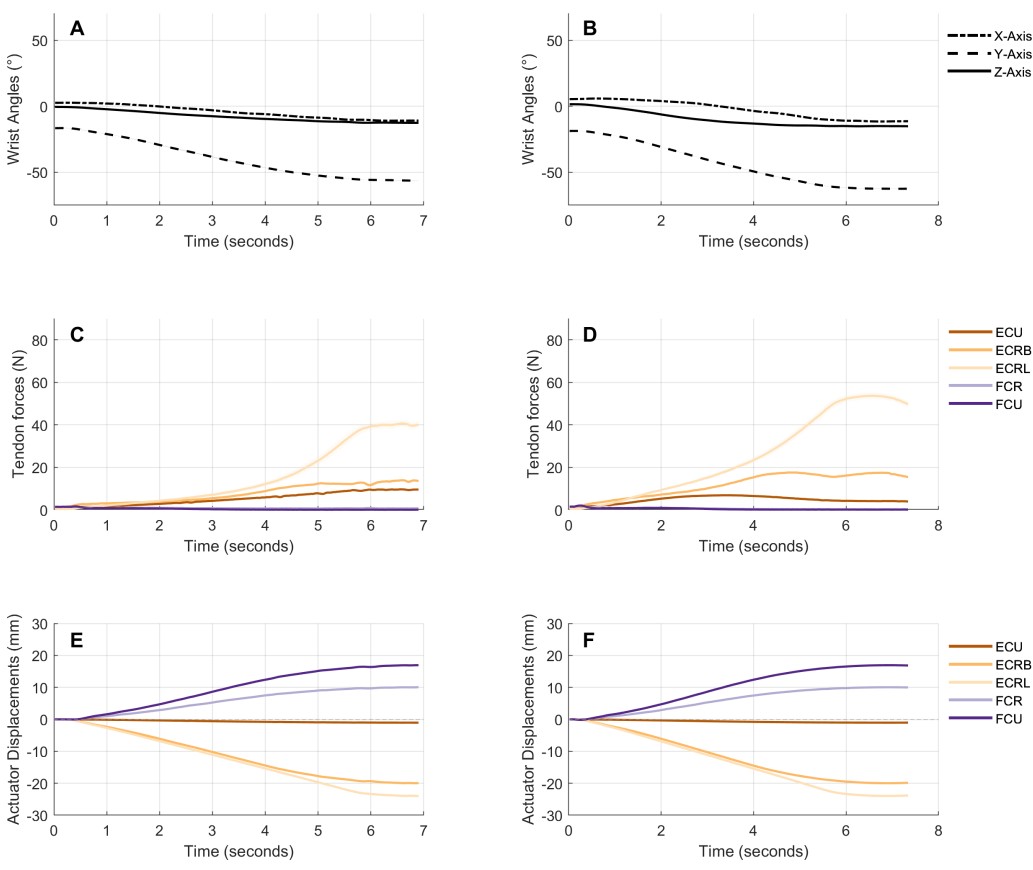

**Figure 3** **Wrist angles, tendon forces and actuator displacements during the average radial extension trial from one representative specimen before and after a total wrist replacement.** (A & B) Angles (°) of the third metacarpal with respect to the radius, represented by rotations about the X (pronation/supination), Y (flexion/extension) and Z (radioulnar deviation) axes of the radial coordinate system. (C &D) Forces (N) applied to the ECU, ECRB, ECRL, FCU and FCR tendons over the duration of the motion trial. (E&F) Displacements (mm) of the five linear actuators connected to the tendons over the duration of the motion trial. The mean angle/force/displacement is plotted at every time point. Standard deviations are represented as a shaded region around the means ($n = 5$).

selections of ADLs used in each study. Nevertheless, wrist angles achieved by our simulator fall towards the upper limit of these fROM ranges (Table 2). It should therefore be possible to determine whether a cadaveric surgery has affected the ability of the wrist to achieve a fROM in future studies using the simulator. For the six wrists included in this pilot study, the total wrist replacements produced similar flexion, extension and radioulnar deviation ranges to the anatomical wrist (Table 2).

In addition to the clinically relevant FE and RUD planes, our motion simulator is also able to produce DTM and circumduction (Figs. 2C & 2D). These motion paths were chosen for their functional relevance. DTM has long been identified as a commonly used plane in ADLs (*Moritomo et al., 2007*), while circumduction encapsulates a wide range of coupled wrist angles (*Li et al., 2005*). The simulator produced RE and UF planes angled at means of 20.4° and 26.2° from the sagittal plane (Fig. 2C). The angle of DTM can be variable
**Table 3  Mean inter-trial RMSEs (±standard deviation) of wrist angle (°), tendon force (N) and actuator displacement (mm) profiles.** All RMSEs represent the axis/tendon/actuator with the maximum error ($n = 6$).

| | Root mean square error (RMSE) | | | | |
|---|---|---|---|---|---|
| | Anatomical angles | Motec angles | Anatomical forces | Motec forces | Actuator displacements |
| Flexion | 0.67 ± 0.56 | 0.30 ± 0.29 | 0.61 ± 0.34 | 0.27 ± 0.21 | 0.25 ± 0.25 |
| Extension | 0.49 ± 0.29 | 0.68 ± 0.99 | 0.9 ± 0.39 | 0.78 ± 0.46 | 0.07 ± 0.05 |
| Radial deviation | 0.46 ± 0.49 | 0.35 ± 0.19 | 0.92 ± 0.3 | 1.41 ± 0.9 | 0.11 ± 0.07 |
| Ulnar deviation | 0.27 ± 0.21 | 0.20 ± 0.10 | 0.38 ± 0.21 | 0.34 ± 0.22 | 0.1 ± 0.14 |
| Radial extension | 0.46 ± 0.36 | 0.43 ± 0.41 | 1.24 ± 0.55 | 0.91 ± 0.43 | 0.12 ± 0.05 |
| Ulnar flexion | 0.44 ± 0.33 | 0.31 ± 0.21 | 0.36 ± 0.23 | 0.23 ± 0.2 | 0.09 ± 0.03 |
| Circumduction | 0.37 ± 0.21 | 0.50 ± 0.31 | 0.98 ± 0.7 | 1.3 ± 0.9 | 0.09 ± 0.04 |

**Table 4  Mean inter-specimen RMSEs (±standard deviation) of wrist angle (°), tendon force (N) and actuator displacement (mm) profiles.** All RMSEs represent the axis/tendon/actuator with the maximum error ($n = 6$).

| | Root mean square error (RMSE) | | | | |
|---|---|---|---|---|---|
| | Anatomical angles | Motec angles | Anatomical forces | Motec forces | Actuator displacements |
| Flexion | 4.67 ± 2.24 | 6.13 ± 3.12 | 3.49 ± 1.45 | 2.54 ± 0.77 | 1.59 ± 0.31 |
| Extension | 5.57 ± 2.47 | 10.68 ± 5.00 | 4.85 ± 2.31 | 6.25 ± 3.78 | 0.69 ± 0.08 |
| Radial deviation | 6.22 ± 3.44 | 7.28 ± 7.21 | 5.94 ± 2.58 | 8.45 ± 4.26 | 0.97 ± 0.62 |
| Ulnar deviation | 5.17 ± 2.95 | 8.60 ± 4.64 | 3.97 ± 2.93 | 7.21 ± 3.59 | 1.92 ± 0.79 |
| Radial extension | 7.85 ± 3.91 | 12.97 ± 6.42 | 9.29 ± 3.61 | 9.14 ± 7.54 | 4.2 ± 2.04 |
| Ulnar flexion | 6.05 ± 2.39 | 6.29 ± 3.49 | 3.05 ± 1.23 | 3.13 ± 0.99 | 1.8 ± 1.52 |
| Circumduction | 7.07 ± 2.19 | 11.51 ± 5.05 | 5.49 ± 2.09 | 8.41 ± 3.33 | 2.81 ± 1.0 |

depending on the purpose of a wrist motion. *Garg et al. (2014)* and *Brigstocke et al. (2014)* measured the angles of DTM during a selection of seven and nine tasks, respectively, and found them to range between of 9–37° and 34–48°. Mean DTM angle during standard dart throwing has been reported as 44° (*Vardakastani et al., 2018*). The angles produced by our simulator are shallower than most reported previously, though do fall within the range reported by *Garg et al. (2014)*. Since the simulator is able to reach a wide range of coupled wrist angles, new actuator displacement profiles can be easily created that produce different angles of DTM.

The simulated circumduction envelopes tended to be broadly elliptical, though with obvious depressions towards the ulnar deviation portion of the motion in some specimens (Fig. 2D). To achieve a more elliptical circumduction motion it was necessary to modify to the actuator displacement profiles. The variation of circumduction ellipse areas across the six subjects was likely due to anatomical differences between the specimens as well as the significant improvements made to the actuator displacement profiles as collection of this pilot data progressed. Despite this, the average ellipse area was within the range calculated by *Gracia-Ibanez et al. (2020)* which encompassed 70–90% of the ADLs measured in their
study, though it is smaller than the maximal circumduction ellipse areas measured by *Singh et al. (2012)* and *Gracia-Ibanez et al. (2020)*.

## Repeatability

Our cadaveric simulator has been designed to isolate wrist angle and tendon force as dependent variables. The independent variables, actuator displacements, initial tendon loading and arm position, are controlled during and between wrist conditions. Wrist angle and tendon force profiles between trials of the same motion and wrist condition were found to be highly repeatable. Indeed, average RMSEs between individual trials and mean profiles were below one degree and two newtons (Table 3). This level of repeatability will enable even small differences in the dependent variables to be detected between wrist conditions. It may also allow for a reduced number of technical replicates, which would be a great benefit given the highly time consuming nature of marker-based XROMM analysis.

Inter-specimen variation in wrist angles and tendon forces was relatively large (Fig. 2 & Table 4). This is likely due to anatomical differences between specimens, differences in initial wrist position and the scaling of displacement profiles. Though inter-specimen variation does not prohibit pre- and post-surgical comparison within an individual wrist, it could be reduced if required by adding wrist position feedback to the simulator.

## Wrist motion simulators

The use of tendon driven wrist motion simulation is an increasingly common method for *in vitro* studies of wrist biomechanics. Previous simulators have used force feedback (*Erhart et al., 2012*) and combinations of wrist position and force feedback (*Werner et al., 1996*; *Iglesias, 2015*; *Shah et al., 2017*; *Razavian et al., 2022*) to control motion. The current study uses actuator displacement control, which is advantageous in that both range of motion and efficiency of motion (the force required to produce a specific movement) can be studied as dependent variables. Additionally, real time wrist angle data is not required. This is important as real time angles cannot be extracted from BPVR data using currently available software and data processing methodologies.

One consequence of using actuator displacement control is relatively high inter-specimen variation between wrist angles (Fig. 2). Similar variability is seen when a simulator is controlled solely by tendon force profiles (*Erhart et al., 2012*). Simulators that use position feedback report wrist angle errors below 2° (*Shah et al., 2017*) and 0.2° (*Razavian et al., 2022*) though this necessitates calculation of real time wrist angles. Moreover, the proposed motion simulator has been designed to achieve consistency and repeatability across different surgeries for the same specimen rather then achieving predefined motions across different specimens.

Force control methods also vary between simulators. This study uses a maximum force boundary informed by the estimated total physiological forces of each muscle. As well as a maximum boundary, several simulators impose a minimum tendon force of 8.9 N to mimic antagonistic muscle forces (*Werner et al., 1996*; *Iglesias, 2015*; *Shah et al., 2017*). As this boundary does not appear to have been founded by *in vivo* data (*Werner et al., 1996*), the extent to which its presence improves the cadaveric model is currently unclear. The

simulator recently described by *Razavian et al. (2022)* does not explicitly define minimum force (*Razavian et al., 2022*). It instead uses a force control mechanism whereby real time tendon forces are calculated based on reference joint moments that in turn are calculated based on measured *vs* reference joint angles. Their resulting antagonistic forces can drop below 8.9 N.

Differences in simulator design doubtless influence the tendon forces used to produce wrist motion. FCR and ECRL forces applied by the current simulator were higher than previously reported, while ECU, ECRB and FCU forces were lower (*Shah et al., 2017*; *Werner et al., 2010*). Decreased ECRB and increased ECRL force is due to the ECRL tendon being the primary instigator of radial deviation in our set up. Decreased ECU force may be due to the lack of abductor pollicis longus (APL), as APL actuation causes increased ECU and decreased FCR and ECRL forces (*Shah et al., 2018*). Unsurprisingly, if eleven muscles of the forearm are actuated, tendon forces tend to be lower than simulators which use 5–6 muscles (*Razavian et al., 2022*).

It is intuitive that the inclusion of more muscles will improve the representation of *in vivo* wrist motion achieved by a cadaveric model. It is also important to note that there is currently no *in vivo* tendon force data with which to make comparisons. Therefore, the main application of our tendon force data will be to compare the efficiency of pre- and post-surgical wrist motion, rather than to draw conclusions relating to the forces applied in life.

## LIMITATIONS

As with any cadaveric simulator, the accuracy with which simulated motion represents *in vivo* conditions cannot be assumed. Most obviously, our simulator only actuates the five major flexor and extensor tendons of the wrist, without considering the ten other muscles that cross the joint. Additionally, our control method does not consider antagonistic muscle forces. It only produces the agonist tendon displacements necessary to simulate a desired wrist motion. These design choices cause simulated wrist motion to deviate from *in vivo* wrist motion. Unlike the control strategy used by *Razavian et al. (2022)*, actuator position control is not reflective of how motor units are recruited to generate forces required for specific movements. This, as well as the fact that we did not simulate motions incorporating external loads, means that the tendon force recordings will not be reflective of *in vivo* muscle forces.

Whether these shortcomings will have a meaningful impact on specific or targeted assessments of wrist kinematics is more difficult to say and will likely vary on a case-by-case basis. During cadaveric wrist motion with and without tendon loading, the difference in carpal bone rotations was determined to be <1.1° and <2.6° for the proximal and distal carpal rows respectively (*Foumani et al., 2010*). These relatively small differences suggest that the influence of tendon loading patterns on carpal bone rotations may be limited. Nevertheless, wrist kinematics resulting from this simulator cannot be considered truly representative of *in vivo* kinematics.

A further consequence of our control method is that wrist angles cannot be quantitatively predefined. While this does allow comparison of wrist angle pre- and post-surgery, it also

means that out of plane motions are difficult to avoid (Table 2). The presence of out of plane motions has the potential to create difficulties as FE and RUD are the most common clinical reference motions.

It should be noted that our simulator positions the wrist with fingers pointing downwards. This was chosen to orient the hand parallel to the gravity vector, thereby minimising its effect. Forearm orientation has been found to influence radioscaphoid joint contact area, as well as lunate rotations at extreme wrist angles (Padmore et al., 2021). It is also likely to alter tendon force distribution. It will therefore be important to consider how forearm orientation may influence results when using this simulator.

Given the limitations discussed, our simulator requires modification before any conclusions can be universally applied to *in vivo* wrist biomechanics. Depending on the research question, it may be beneficial to incorporate antagonistic muscle force and wrist position feedback. The latter would require optical motion capture or wearable sensors (Engstrand et al., 2021) as well as BPVR to capture real time angles.

## CONCLUSIONS

This article has presented an active motion simulator capable of producing FE, RUD, DTM and circumduction in cadaveric wrists. High inter-trial repeatability of wrist angles and tendon forces has been demonstrated, highlighting the potential for meaningful comparison of these variables between pre- and post-surgical wrists. The simulator is also compatible with BPVR techniques which will permit the use of marker-based XROMM to compare parameters such as carpal kinematics and joint centre of rotation. However, the application of this simulator in its current form is limited to identifying potential mechanical effects of motion preserving wrist surgeries. To be a more useful tool in the study of wrist biomechanics, representation of the *in vivo* wrist should be improved.

## ACKNOWLEDGEMENTS

For assembling the simulator we would like to acknowledge the School of Engineering technical team at the University of Liverpool. We would like to thank Ms Harriet Julian who assisted the clinical author in the surgical preparation of cadaveric specimens. We also thank Karen Collins, Steph Hanna and Melissa McAdam for obtaining CT scans.

### Funding

This work was funded by the Engineering & Physical Sciences Research Council (Grant no. 2439558), Swemac, the University of Liverpool and Medical Research Council (MRC) and Versus Arthritis as part of the MRC Versus Arthritis Centre for Integrated Research into Musculoskeletal Ageing (CIMA) (Grant no. MR/P020941/1). The funders had no role in study design, data collection and analysis, decision to publish, or preparation of the manuscript.

## Grant Disclosures

The following grant information was disclosed by the authors:

The Engineering & Physical Sciences Research Council, Swemac: 2439558.

The University of Liverpool and Medical Research Council (MRC) and Versus Arthritis as part of the MRC Versus Arthritis Centre for Integrated Research into Musculoskeletal Ageing (CIMA): MR/P020941/1.

## Competing Interests

The authors declare there are no competing interests.

## Author Contributions

- Joanna Glanville conceived and designed the experiments, performed the experiments, analyzed the data, prepared figures and/or tables, authored or reviewed drafts of the article, and approved the final draft.
- Karl T. Bates conceived and designed the experiments, performed the experiments, authored or reviewed drafts of the article, and approved the final draft.
- Daniel Brown conceived and designed the experiments, performed the experiments, authored or reviewed drafts of the article, and approved the final draft.
- Daniel Potts performed the experiments, authored or reviewed drafts of the article, designed and built the Wrist Motion Simulator, and approved the final draft.
- John Curran performed the experiments, authored or reviewed drafts of the article, designed and built the Wrist Motion Simulator, and approved the final draft.
- Sebastiano Fichera conceived and designed the experiments, performed the experiments, authored or reviewed drafts of the article, and approved the final draft.

## Human Ethics

The following information was supplied relating to ethical approvals (i.e., approving body and any reference numbers):

University of Liverpool Central University Research Ethics Committee granted approval for this study (Reference 8009).

## Data Availability

The raw biplanar data, maya animations and the MATLAB scripts used to analyse the data and produce the tables and figures are available in the University of Liverpool DataCat repository:

https://doi.org/10.17638/datacat.liverpool.ac.uk/2509.

## Supplemental Information

Supplemental information for this article can be found online at http://dx.doi.org/10.7717/peerj.17179#supplemental-information.

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
