# Peer review of "Evaluation of a cadaveric wrist motion simulator using marker-based X-ray reconstruction of moving morphology"

_PeerJ, doi:10.7717/peerj.17179_

## Round 0.1 · original submission · Major Revisions

Reviewer 2 attached a separate file with comments. Please make sure to address those comments.

In addition to the reviewer comments, I have some additional comments.

1. In the final paragraph of the Introduction, please change "principle criteria" to "principal criteria". "Principle" is never used as an adjective.

2. Reviewer 1 challenged the conclusion that the system can be used to determine the effect of surgeries on wrist biomechanics. To address, this, I would suggest to present the study as an "evaluation" rather than "validation". Clearly the tests demonstrate repeatability and accuracy, but relevance to in vivo wrist biomechanics is a separate issue which should be acknowledged as requiring further work.

3. The main criterion for performance is the repeatability of the force and motion trajectories. There are many other methods that would be equally repeatable, including moving the bones directly, but those would not necessarily be "valid" for the research questions that are envisioned.

4. The tendon forces are quite low, and do not seem to represent tasks with external loads resisting the motion, or tasks where there is antagonistic co-contraction. This is a limitation that should be acknowledged. As stated by the authors in the Introduction as a motivation for the work, joint kinematics and muscle moment arms may be load dependent.

Reviewer 1 ·

Basic reporting

The authors developed a muscle-based actuation system for wrist motion simulation. They placed this simulation in a biplane flouro setting to capture 3D kinematics of the bones of the wrist. This work is important and needed in the biomechanics field.

Experimental design

The manuscript accurately describes the steps that were taken and the data/results that were found. A fundamental concern with the work is that the conclusion “The combination of highly repeatable cadaveric motion simulation with BPVR and XROMM can be used to determine the effects of different surgeries on wrist kinematics.” is undercut by the methodology. The simulation approach of prescribing muscle length profiles rather than controlling forces or wrist kinematics feels like quite a deviation from the in vivo environment. Concluding that this simulator is able to “determine the effects of different surgeries on wrist kinematics” does not seem correct. Preventing overstatement of the conclusion would help paint a more accurate manuscript for the readers. For example, “The combination of specimen specific highly repeatable cadaveric muscle motion simulation with BPVR and XROMM can be used to determine some potential effects of different surgeries on wrist kinematics.”

In the abstract, only intra-specimen repeatability was presented. Considering the title of the manuscript, I would have expected inter-specimen variability to also be described as a metric of validation. Remember that both positive and negative data can result during a validation and the system can be validated for intra-specimen, but not inter-specimen variability. “High intra-specimen repeatability of motion trials was achieved. Root mean square errors between individual trials and average wrist angle and tendon force profiles were below 1° and 1 N respectively.” Figure 2 shows quite a bit of variability. In addition, this statement in line 165 is quite problematic considering the control technique. “If necessary, displacement profiles were scaled in order to achieve a more optimal or different magnitude of wrist motion.” This suggests that the operators modified the position control trajectories differently for each specimen to achieve a specific wrist motion. Even then, Figure 2 variability is still quite large across specimens. The results also do not list force variability across specimens. This can be seen in the supplemental data, but that only underscores my point about simulated muscle forces being highly variable. For example, presenting the data from Table 3 across all specimens would be of additional benefit to the reader.

There is also a final fundamental challenge with this manuscript in that it does not present a novel mechatronic solution. The introduction lays out descriptions of the other simulators that have been developed and, except for placing the simulator in a biplane flouro motion capture environment, the other simulators have the same, or better features. If one of the benefits of the system is that it is lower cost, then this should be stated. If one of the benefits is that it is faster, then this should be stated. I am not seeing evidence that either are the case, and it appears that the science is being sacrificed because of the limitations of the mechatronic technology.

Line-by-line:

Figure 1: The model wrist shown in the CAD model and the picture is not representative of the cadaveric validation study. Can these be updated with models of a human wrist, or pictures from the actual experiment? I think this will help the reader better grasp what was done.

Line 103: Simulator hardware: Is there a reason that a rotary encoder or resolver was not used on the motor, rather than the magnetic tape and linear encoder? Obviously, the design choice presented works, but could be problematic long term in a cadaveric environment.

Line 125: “This method of control allows the same actuator displacements to be applied to both pre- and post- surgical wrists.” This statement, while true, may not be a proper simulation of the in vivo environment. Muscles are more force controlled actuators than they are position controlled actuators. Proprioception assists with closing the loop on position, but this concept supports a simulator design where the actuators are adjusted to meet wrist motions, not specific muscle lengths.

Line 140: The use of suture tape to connect the tendons to the machine is concerning, especially because the position control nature of the simulator. Differences in the taping process, and stiffness/security of the connection can influence the resulting motion and loads on the wrist.

Line 453: “This paper has presented an active motion simulator capable of producing repeatable FE, RUD, DTM and circumduction in cadaveric wrists.” I am not sure this statement is supported by the manuscript. Inter-specimen variability for these metrics would not, in my estimation, be considered repeatable. The supplemental data makes it clear that the individual actuator force and position profiles were highly variable amongst specimens. In addition, even with the manual interventions to the profiles outlined in line 165 the specimen-to-specimen variability shown in Figure 2 was highly variable. So, we cannot even conclude that the trained operator is able to close the loop for the simulator to produce similar specimen-to-specimen results.

Validity of the findings

I understand that going back, redesigning the system to include closing the loop on forces and wrist kinematics, running all the experiments again, and rewriting this manuscript is not a practical suggestion for many reasons. It is also a reasonable argument to make that the current state of the system should be published as a point of reference. It is a launching point for the authors to add more capability to their lab infrastructure in the future. However, the way the current manuscript reads is with more confidence in a ‘validated’ system than is merited. I would not consider the results to support that this system is now valid for future wrist biomechanics research. Instead, I would point out the elements that were good (run to run repeatability within a single specimen, simulation and measurement of wrist motion in a biplane flouro environment) and ALSO point out the things that weren’t (listed above). Then conclude that while this system has some benefits, future work will need to be done to address issue x, y, and z, before it can be considered ‘validated’ for wrist biomechanics research. The title of the manuscript can be less focused on validation and more on 'description' or 'demonstration' of a system...

Reviewer 2 ·

Basic reporting

Language is clear. Literature review is sufficient. Figures and tables are understandable. The manuscript could be made more concise.

Experimental design

The purpose of the design, as it pertains to making improvements over previous designs, can be emphasized more clearly.

Validity of the findings

The findings could be strengthened with the inclusion of statistical comparisons between the pre- and post-surgical conditions.

Annotated reviews are not available for download in order to protect the identity of reviewers who chose to remain anonymous.

---

## Round 0.2 · Minor Revisions

This is being returned to you per prior discussion of the changes you wished to make.

Reviewer 1 ·

Basic reporting

Requested improvements were made

Experimental design

Requested improvements were made

Validity of the findings

Requested improvements were made

Reviewer 2 ·

Basic reporting

Language is clear. Literature review is sufficient. All figures and tables are understandable and have been updated to include the suggested improvements.

Experimental design

The description and discussion of the experimental design has been updated and improved in comparison to the previous submission.

Validity of the findings

The findings were strengthened with the addition of statistical comparisons.

Additional comments

The authors have adequately addressed all of the comments. The added analyses, figure updates and text edits have improved the manuscript.

---

## Round 0.3 · accepted · Accept

Thank you for resubmitting the manuscript with additional changes. The reporting of anatomical wrist angles, and the use of a metacarpal coordinate system is indeed an improvement and will increase the relevance of the work. I have reviewed the changes and can confirm that none of the changes require input from the reviewers. The manuscript is ready for publication.